# SELECTIVITY METRICS CAN OVERESTIMATE THE SELECTIVITY OF UNITS: A CASE STUDY ON ALEXNET

## ABSTRACT

Various methods of measuring unit selectivity have been developed in order to understand the representations learned by neural networks (NNs). Here we undertake a comparison of four such measures on AlexNet, namely, localist selectivity Bowers et al. (2014), precision (Zhou et al., 2015), class-conditional mean activity selectivity CCMAS; Morcos et al. (2018), and a new measure called top-class selectivity. In contrast with previous work on recurrent neural networks (RNNs), we fail to find any 100% selective 'localist units' in AlexNet, and demonstrate that the precision and CCMAS measures provide a much higher level of selectivity than is warranted, with the most selective hidden units only responding strongly to a small minority of images from within a category. We also generated activation maximization (AM) images that maximally activated individual units and found that under (5%) of units in fc6 and conv5 produced interpretable images of objects, whereas fc8 produced over 50% interpretable images. Furthermore, the interpretable images in the hidden layers were not associated with highly selective units. These findings highlight the problem with current selectivity measures and show that new measures are required in order to provide a better assessment of learned representations in NNs. We also consider why localist representations are learned in RNNs and not AlexNet.

## 1  INTRODUCTION

Although previously seen as black boxes, there have been recent attempts to understand how neural networks (NNs) work by analyzing hidden units one at a time using various measures such as localist selectivity (Bowers et al., 2014), class-conditional mean activity selectivity (CCMAS) (Morcos et al., 2018), precision (Zhou et al., 2015), and activation maximization (AM) (Erhan et al., 2009b). These measures are defined below, and they all provide evidence that some units respond selectively to categories under some conditions. For example, Bowers et al. (2014; 2016) found localist letter and word representations in recurrent networks (RNNs) trained on short-term memory tests, and (Zhou et al., 2015; 2018) reported object detectors in convolutional neural networks (CNNs) trained on ImageNet.

Our goal here is to directly compare different measures of object selectivity on a common network trained on a single task. We chose AlexNet (Krizhevsky et al. (2012)) because it is a well-studied CNN and many authors have reported high levels of selectivity in its hidden layers via both quantitative (Zhou et al., 2018; 2015) and qualitative methods using Activation Maximization (AM) images (Nguyen et al., 2017; Yosinski et al., 2015; Simonyan et al., 2013). Our main findings are:

1. The precision and CCMAS are misleading measures that overestimate selectivity.

2. There are no localist 'grandmother cell' representations in AlexNet, in contrast with the localist representations learned in some RNNs.

3. Units with interpretable AM images do not necessarily correspond to highly selective representations.

4. New selectivity measures are required to provide a better assessment of the learned hidden representations in NNs.

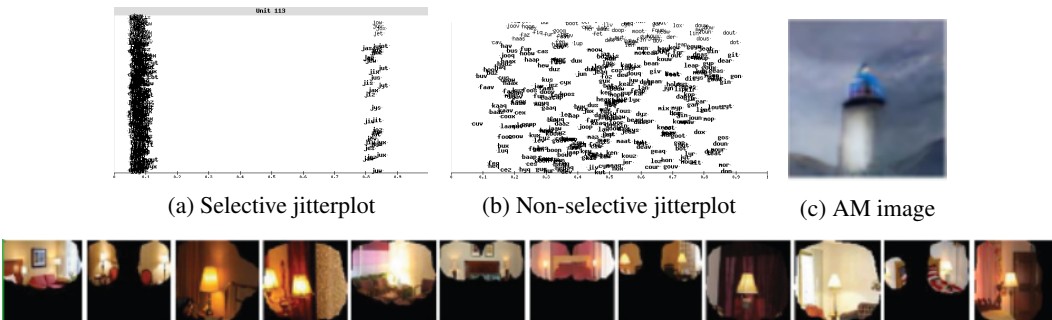

(a) Selective jitterplot (b) Non-selective jitterplot (c) AM image

(d) Example images that strongly activate a high-precision lamp-detector unit

Figure 1: Examples of selectivity measures used. (a) Jitterplot of unit 113 in an RNN under the superposition constraint selective the letter 'j' (b) Jitterplot of a non-selective unit 160 found when RNN trained on words one-at-a-time; from Bowers et al. (2016). (c) Activation maximization (AM) image of a unit in `conv5` of AlexNet that looks like a lighthouse; from Nguyen et al. (2016). (d) Highest activation images for a 'lamp' detector with 84% precision in layer `pool5` of AlexNet; from Zhou et al. (2015).

Bowers et al. (2014; 2016) assessed the selectivity of hidden units in recurrent NNs using networks similar to those developed by Botvinick & Plaut (2006) designed to explain human performance on short-term memory tests. They reported many 'localist' units that are 100% selective for specific letters or words, where all members of the selective category were more active than and disjoint from all non-members, as can be shown in jitterplots (Berkeley et al., 1995) (see Fig. 1a for an example of a unit selective to the letter 'j'). These localist representations were compared to 'grandmother cells' as discussed in neuroscience (Bowers, 2017a). Bowers et al. (2014) argued that the network learned these representations in order to co-activate multiple letters or words at the same time in short-term memory without producing ambiguous blends of overlapping distributed patterns (the so-called 'superposition catastrophe'). Consistent with this hypothesis, localist units did not emerge when the model was trained on letters or words one-at-a-time (a condition in which the model did not need to overcome the superposition catastrophe (Bowers et al., 2014)). (see Fig. 1b for an example of a non-selective unit trained under this latter condition)

In parallel, researchers (Zhou et al. 2015; Morcos et al. 2018; Zeiler & Fergus 2014; Erhan et al. 2009a, for a review see Bowers (2017a)) reported selective units in the hidden layers of various CNNs, including AlexNet (Krizhevsky et al., 2012), trained to classify images into one of multiple categories. For example, Zhou et al. (2015) assessed the selectivity of units in the `pool5` layer of two CNNs trained to classify images into 1000 objects and 205 scene categories, respectively. They reported multiple 'object detectors' (as defined in the method section) in both networks, Similarly, Morcos et al. (2018) reported that CNNs trained on CIFAR-10 and ImageNet learned many highly selective hidden units, with CCMAS scores often approaching the maximum of 1.0. Again, these results suggest high-levels of selectivity in CNNs.

Note that these later studies show that selective representations develop in CNNs trained to classify images one-at-a-time. This appears to be inconsistent with the Bowers et al. (2016) who (a) failed to obtain selective representations for letters or words under these conditions (see Fig. 1b); and (b) it suggests that there are additional pressures for CNNs to learn selective representations above and beyond the challenge of overcoming the superposition catastrophe. However, the measures of selectivity that have been applied across studies are different, and accordingly, it is difficult to directly compare results.

In order to directly compare and have a better understanding of the different selectivity measures we assessed (1) localist, (2) precision, and (3) CCMAS selectivity on the prob, fc8, fc7, fc6, and conv5 layers of AlexNet. We also introduce a new measure called top-class selectivity, and show that the precision and CCMAS measures provide much higher estimates of object selectivity compared to other measures. Importantly, we do not find any localist 'grandmother cell' representations in the hidden layers of AlexNet, consistent with the hypothesis that the superposition catastrophe provides a pressure to learn more selective representations Bowers et al. (2014; 2016).

In addition, we compared these selectivity measures to a state-of-the-art activation maximization (AM) method for visualizing single-unit representations in CNNs (Nguyen et al., 2017). AM images are generated to strongly activate individual units, and some of them are interpretable by humans (e.g., a generated image that looks like a lighthouse, see Fig. 1c). For the first time, we systematically evaluated the interpretability of the AM images in an on-line experiment and compare these ratings with the selectivity measures for corresponding units. We show that hidden units with interpretable AM images are not highly selective.

## 2 METHODS

**Networks and Datasets** All ∼1.2M photos from ImageNet2010 (Deng et al. 2009) were cropped to $277 \times 277$ pixels and classified by the pre-trained AlexNet CNN (Krizhevsky et al. 2012) shipped with Caffe (Jia et al. 2014), resulting in 721,536 correctly classified images. Once classified, the images are not re-cropped nor subject to any changes. In Caffe, the softmax operation (Denker & leCun 1991) is applied at the 'prob'(ability) output layer that contains 1000 units (one for each class). We analyzed these prob units, the fully connected (fc) layers: fc8 (1000 units) that encodes the outputs prior to the softmax operation, fc6 and fc7 (4096 units), and the top convolutional layer conv5 which has 256 filters. We only recorded the activations of correctly classified images, and saved them in an activation table so the activations could be probed without re-evaluating the images each time. The activation files are stored in .h5 format and can be retrieved at `http://anonymizedForReview`. We selected 233 conv5, 2738 fc6, 2239 fc7, 911 fc8, and 954 prob units for analysis.

**Localist selectivity** Here we define a unit to be localist for class $A$ if the set of activations for class $A$ was disjoint with those of $\neg A$. A unit is selectively 'on' if $\{A\} > \{\neg A\}$ (i.e. all images in $A$ have higher activations than those not in $A$) and selectively 'off' if $\{A\} < \{\neg A\}$. Localist selectivity is easily depicted with jitterplots in which a scatter plot for each unit is generated (see Figs. 3a and 4a, b). Each point in a plot corresponds to a unit's activation in response to a single image, and only correctly classified images are plotted (if an image has been misclassified we cannot use its label to elucidate what the unit responds to). The level of activations is coded along the $x$-axis, and an arbitrary value is assigned to each point on the $y$-axis (they are jittered). When generating jitterplots for the conv5 layer we plotted the highest level of activation across each filter for each image.

**Top-Class selectivity** Top-class selectivity is closely related to localist selectivity except that it provides a continuous rather than discrete measure. We counted the number of images from the same class that were more active than all images from all other classes (what we called the top cluster size) and divided the cluster size by the total number of correctly identified images from this class. 100% top-class selectivity is equivalent to a localist representation.

**Precision** The precision method of finding object detectors (Zhou et al., 2015; 2018) involves identifying a small subset of images that most strongly activate a unit (the number of images in the most strongly activated subset differ across papers) and then identifying the critical part of these images that are responsible for driving the unit. Zhou et al. 2015 took the 60 images that activated a unit the most strongly and asked independent raters to interpret the critical image patches. Zhou et al. (2015) developed a precision metric that assessed the percentage of the 60 images that raters judged to depict the same class of object (e.g., if 50 of the 60 images were labeled as 'lamp', the unit would have a precision index of 50/60 or 83%; see Fig. 1d). Object detectors were defined as units with a precision > 75%: they reported multiple such detectors. Here we approximate this approach by considering the 100 images that most strongly activate a given unit and assess the highest percentage of images from a given output class (e.g., if 75 of the top 100 images are all examples of a class 'lighthouse' then we consider the unit to be a 'lighthouse' object detector with a precision of 75%).

**CCMAS** Morcos et al. (2018) introduced a selectivity index based on the 'class-conditional mean activation' selectivity (CCMAS). The CCMAS for class $A$ compares the mean activation of all images in class $A$, $\mu_A$, with the mean activation of all images not in class $A$, $\mu_{\neg A}$, and is given by: $(\mu_A - \mu_{\neg A}) / (\mu_A + \mu_{\neg A})$. Morcos et al. (2018) states that this metric should vary within [0,1], with 0 meaning that a unit's average activity was identical for all classes, and 1 meaning that a unit was only active for inputs of a single class. Here, we assessed class selectivity for the highest mean activation class (CCMAS) as well as for the class with the second highest mean activation $\mu_A$ (what

we call CCMAS_2) in order to assess the extent to which CCMAS reflects the selectivity to one class.

**Activation Maximization** We harnessed an activation maximization method called Plug & Play Generative Networks (Nguyen et al., 2017) in which an image generator network was used to generate images (hereafter, AM images) that highly activate a unit. We generated 100 separate images that maximally activated each unit in the conv5, fc6 and fc8 layers of AlexNet and displayed them in a grid format (see Appendix Figs. A5, A6 & A7). We then asked 333 participants to judge whether they could identify any repeating objects, animals, or places in images after receiving some practice trials (see Appendix Fig. A1 for an example). Participants were recruited using Prolific (pro; Palan & Schitter, 2018), with the experiment run online using Gorilla (gor). More details of the experiment can be found in the Appendix A1 and an example experiment for readers to try is at: exp.

## 3 RESULTS

### 3.1 COMPARISON OF SELECTIVITY MEASURES.

The mean top-class, precision, and CCMAS selectivities across the conv5, fc6, fc7, fc8, and prob layers are displayed in Fig. 2a–c. We did not plot localist selectivity as there were no localist 'grandmother units' at any level, including the prob layer. The first point to note is that the top-class, precision, and CCMAS measures all increased in the higher layers, showing that they capture degrees of selectivity ignored by the localist measure. The second and perhaps the most striking finding is that top-class selectivity was extremely low across the hidden layers, with means below 0.25% in the the conv5, fc6, and fc7 layers. Third, the different measures provided very different estimates of selectivity. In contrast with top-class selectivity, the mean precision scores are over an order of magnitude larger in the hidden layers of network, with average precision scores of 9.6%, 12.1%, and 15.4% in layers conv5, fc6, and fc7, respectively. Similarly, the CCMAS measure suggests a much higher level of selectivity than top-class selectivity, with mean scores of .49, .84, and .85 in the conv5, fc6, and fc7 layers, respectively.

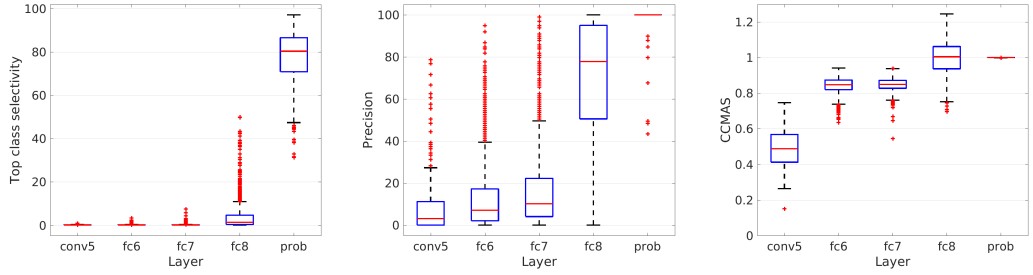

Figure 2: Selectivity measures across different layers of AlexNet. Left: top-class selectivity. Middle: precision 100 (the percentage of the top 100 images which are members of the top class). Right: Class-conditional mean activity selectivity (CCMAS).

The discrepancy between precision and CCMAS on the one hand, and top-class selectivity on the other, is even more striking when precision and CCMAS scores are high, as shown in Table 1. For example, we find 4.5% of units in layer fc7 have a precision of over 50% (that is 184 units) and over 80% of units in fc7 have a CCMAS measure of over .92. At the same time, only 0.1% of units have a top-class selectivity over 5%. Indeed, there are no units with a top-class selectivity over 5% in fc6 or conv5. Based on the precision and CCMAS measures it appears that AlexNet has learned some highly selective representations for objects, but according to localist and top-class selectivity, there is no evidence for this conclusion.

This discrepancy becomes more striking still when you consider the units with the highest precision and CCMAS scores (see Table A1 in the Appendix for examples across multiple layers of AlexNet). To highlight one example, in Fig. 3 we illustrate why the unit fc6.1199 with the highest precision (95%) in fc6 should not be considered a Monarch butterfly detector. Fig. 3a depicts a jitterplot of activations to all accurately identified images, with Monarch butterfly images found across the range of activations. Fig. 3b shows a histogram that plots the distribution of activations for Monarch

Table 1: The percentage of precision, CCMAS and top-class units in each layer at different threshold cut-offs.

| Layer | % of units at various thresholds | | | | | |
|---|---|---|---|---|---|---|
| | **Precision** | | **CCMAS** | | | **Top-Class selectivity** |
| | over 50% | over 75% | over 0.7 | over 0.8 | over 0.9 | over 5% |
| prob | 99.7% | 99.6% | 100% | 100% | 100% | 100% |
| fc8 | 75.4% | 32.8% | 99.9% | 97.2% | 84.5% | 23.6% |
| fc7 | 4.5% | 0.3% | 99.9% | 92.1% | 5.0% | 0.1% |
| fc6 | 3.0% | 0.1% | 99.7% | 86.7% | 6.7% | 0% |
| conv5 | 4.7% | 0% | 3.4% | 0% | 0% | 0% |

butterflies. By far the most common activation to correctly identified Monarch butterflies is 0 and the mean is 39.2±0.6. Figures 3c–e displays example images with 0 (c), mean (d) and maximal (e) activations, and all are identifiable as Monarch butterflies. Thus, classifying this unit as a Monarch butterfly detector is misleading.

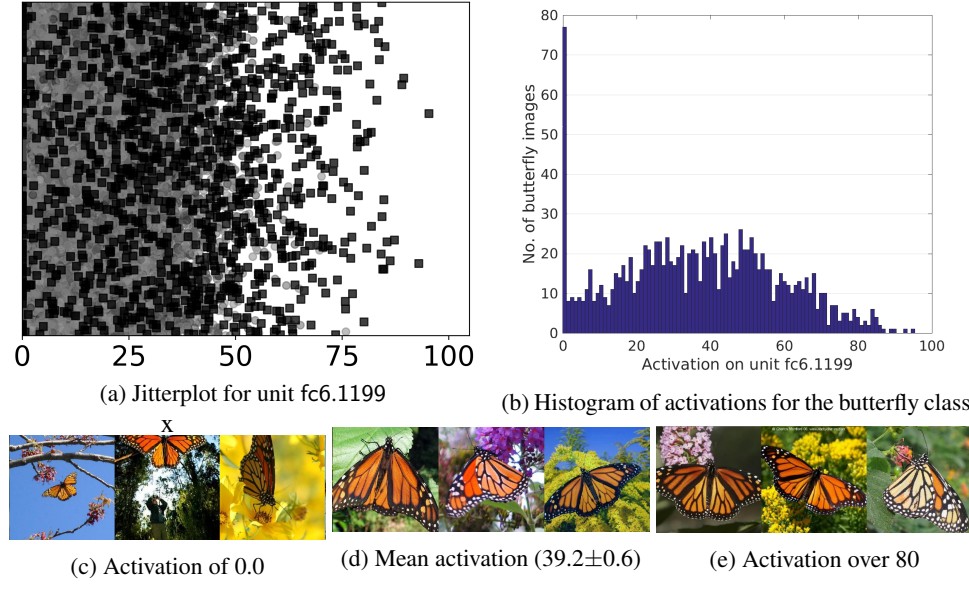

(a) Jitterplot for unit fc6.1199      (b) Histogram of activations for the butterfly class.

(c) Activation of 0.0      (d) Mean activation (39.2±0.6)      (e) Activation over 80

Figure 3: Data for unit fc6.1199. (a) activation jitterplot black squares: Monarch butterfly images; grey circles: all other classes. (b) histogram of activations of Monarch butterflies, c-e example ImageNet images with activations of 0, the mean, and the maximum of the range. Unit fc6.1199 has a precision of 95% over the top 100 images (98.3% over the top 60) and is thus classified as a butterfly detector, yet there are Monarch butterfly images covering the whole range of values, with 72 images (5.8% of the total) having an activation of 0.0.

Another surprising result is that we did not obtain any 100% top-class selectivity units (localist units) in the prob layer of AlexNet. Rather, the mean top-class selectivity was ∼80% in the prob layer, and only ∼5% in fc8 (prior to the softmax being applied). Figs. 4a,b depict the pattern of activation for units fc8.11 and prob.11 that are examples of the most top-class selective units in these layers (responding to images of 'goldfinch' birds with top-class selectivity measures of 8.4% and 95.2%, respectively). Clearly these units do respond much more selectively than the most selective units in earlier layers (as in Fig. 3), and at the same time, the jitterplots show why we did not observe any localist units (a few 'goldfinch' images were less active than a few images from other categories).

These jitterplots also show that top-class and localist selectivity provide somewhat misleading estimates of selectivity as well. Consider Fig. 4a that depicts a substantial overlap between goldfinch and non-goldfinch activations on unit fc8.11. The 8.4% top-class selectivity score captures the se-

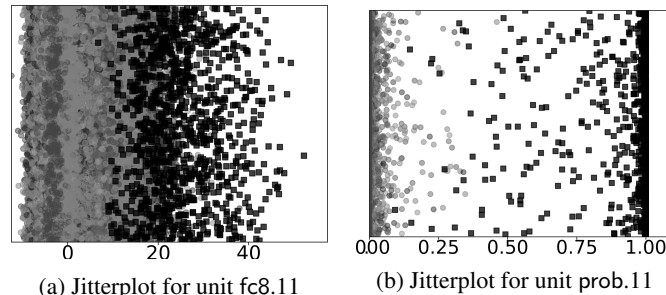

(a) Jitterplot for unit fc8.11     (b) Jitterplot for unit prob.11

Figure 4: Example data from the fc8 and prob layers. (a) jitterplot activations for unit fc8.11 that has a top-class selectivity of 8.4%. (b) jitterplot activations for prob.11 (i.e. post-softmax) that has top-class selectivity of 95.2%. Activations for the 'ground truth' class 'goldfinch' are shown as black squares, all other classes are shown as colored circles.

lectivity for the most highly active goldfinch images, but it is blind to the fact that almost goldfinch images have a reasonably high level of activation (more than most non-goldfinch images). The problem with localist selectivity is highlighted in Fig. 4b that shows that the measure misses all but the most extreme version of selectivity. Together, these findings suggest that additional selectivity measures are required to better characterize the learned representations in NNs: precision and CCMAS measures strongly overestimate the degree of selectivity, and localist and top-class selectivity provide either a too strict or too narrow a measure of selectivity.

## 3.2 Additional problems with the CCMAS measure.

The main problem with the precision and CCMAS measures is that they provide misleadingly high estimates of selectivity, but the CCMAS measure has some additional limitations. The first point to note is that contrary to Morcos et al. (2018), the CCMAS measure can go above 1 if $\mu_{\neg A}$ is negative, and this happens in the fc8 layer of AlexNet (see Fig. 4a) because there is no ReLU transformations in layer fc8 (as opposed to layers fc6 & fc7). This is a minor issue that can be fixed by normalizing the range of activations, but it explains why some of our CCMAS scores are above 1. More importantly, if the CCMAS provided a good measure of a unit's class selectivity then one should expect that a very high measure of selectivity for one class would imply that the unit is not highly selective for other classes. This is not the case as shown in Fig. 5a-c where CCMAS are compared to the CCMAS_2 that assesses unit selectivity to the category with the second highest mean activation. For example, unit fc7.0 has a CCMAS of .813 for the class 'maypole', but the class with the second highest mean activation, 'chainsaw' has a CCMAS of .808 (and neither of these is the top-class which is 'orangutan' and has a precision of 14%).

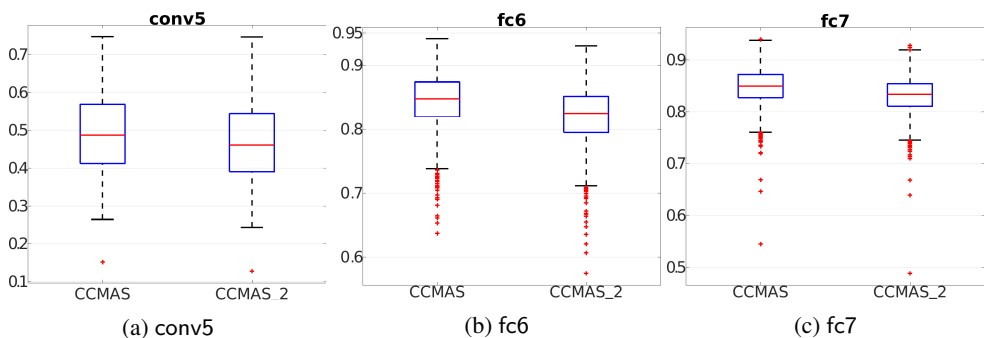

(a) conv5     (b) fc6     (c) fc7

Figure 5: Example of where the CCMAS does not match intuitive understandings of selectivity. Experimental data: (a-c) the CCMAS scores for the class with the highest (CCMAS) and second highest mean activation (CCMAS_2) for all units across layer conv5 (a), fc6 (b), and fc7 (c).

It is also important to note that the CCMAS measure of selectivity is measuring something quite different to alternative measures. For example, the percentage of conv5, fc6 and fc8 units in which

top-class and CCMAS are selective to the same class as follows: 0%, 9.8%, and 83.5% (0.1% being chance). To highlight how discrepant the different measures can be, we have generated some artificial datasets depicted in Fig. A4 in the Appendix that show that CCMAS scores can be much higher or lower than top-class or localist selectivity scores. Indeed, the figure shows that the CCMAS measure can give a low selectivity score to a localist representation. Part of the problem is that the CCMAS measure compares the mean of the selected class with the mean of the unselected classes, but these distributions are not normal (in fact activations of all classes follows an exponential distribution), and thus a comparison between means is not an appropriate measure: see Figures A2 and A3 in Appendix section A3).

### 3.3 Human interpretation of generated images from across AlexNet

The results of the behavioral experiment in which humans rated AM images are reported in Table 2. Consistent with past research, the generated images in the output fc8 layer were often interpreted as objects, and when they were given a consistent interpretation, they almost always (95.4%) correspond to the trained category. By contrast, less than 5% of units in conv5 or fc6 were associated with consistently interpretable images, and as can be seen in Table 2, the interpretations only weakly matched the category with the highest top-class or CCMAS selectivity. The frequency with which objects were seen by the participants was similar in layers conv5 and fc6 layers and increased in fc8, consistent with the top-class and and precision measures of selectivity.

Apart from showing that there are few interpretable units in the hidden layers of AlexNet, our findings show that the interpretability of images does not imply a high level of selectivity. That is, we know from Sec. 3.1 that the maximum top-class selectivity for the hidden units is well under 10% (Fig. 2), and accordingly, all the consistently interpretable units had selectivities less than this. Indeed, in most cases, the top-class selectivity of the interpretable units is well under 1%. To briefly illustrate the types of images that participants rated as objects or non-objects, Fig. 6a–c depicts three AM images from units in the conv5, fc6, and fc8 layers of AlexNet that were interpreted consistently with the top-class selectivity measure, and Fig. 6d-e depicts corresponding uninterpretable images. Additional images can be found in the Appendix Figs. A5 and A6.

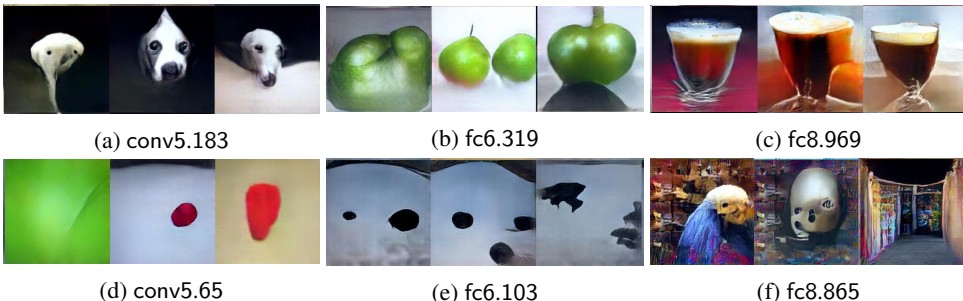

(a) conv5.183          (b) fc6.319          (c) fc8.969

(d) conv5.65          (e) fc6.103          (f) fc8.865

Figure 6: Example AM images that were either judged by all participants to contain objects (a–c) or judged by all participants to be uninpterpretable as objects (d–e). The human judgement for conv5.183 was 'dogs' and the top-class was 'flat-coated retriever'. For fc6.319 subjects reported 'green peppers' or 'apples' (all classified as the same broad class in our analysis), and the CCMAS and top-class was 'Granny Smith apples'. For fc8.969 humans suggested 'beverage' or 'drink': ground truth class for this unit is 'eggnog'. The ground-truth for fc8.865 is 'toy-store'.

## 4 Discussions and Conclusions

Our central finding is that different measures of activation selectivity support very different conclusions when applied to the same units in AlexNet. In contrast with the precision (Zhou et al. 2015) and CCMAS (Morcos et al. 2018) measures that revealed some highly selective units for objects in layers conv5, fc6, and fc8, we found no localist representations, and the mean top-class selectivity in these layers was well under 1%. These findings are in stark contrast with the many localist 'grandmother cell' representations learned in RNNs Bowers et al. (2014; 2016); Bowers (2017b).

Table 2: Interpretability judgements for AM images. The number of judgments for conv5, fc6 and fc8 were 1332, 10,656 and 5,181, respectively.

| LAYER | % YES RESPONSES | % OF UNITS WITH $\geq 80\%$ YES RESPONSE | AMONG HUMANS | % OVERLAP BETWEEN HUMANS and TOP CLASS | CCMAS CLASS |
|-------|-----------------|-------------------------------------------|--------------|----------------------------------------|-------------|
| conv5 | 21.7% ±1.1% | 4.3% ± 1.3% | 89.5%±5.7% | 34.1%±14.4% | 0% |
| fc6 | 21.0% ±0.4% | 3.1% ± 0.4% | 80.4%±4.1% | 23.3%±5.9% | 18.9% ±5.9% |
| fc8 | 71.2% ±0.6% | 59.3% ±1.6% | 96.5%±0.4% | 95.4%±0.6% | 94.6% ±0.7% |

Not only did the different measures provide very different assessments of selectivity, we found that the precision and CCMAS measures provided highly misleading estimates. For example, a unit with over a 75% precision score for Monarch butterflies had a top-class selectivity of under 5%. Although Zhou et al. (2015) used 75% precision scores as the criterion for 'object detectors', it is inappropriate to call this unit a Monarch butterfly detector given that it did not respond strongly to the majority of Monarch butterfly images (and indeed, the modal response was 0; see Fig. 3). This discrepancy between precision and top-class selectivity was widespread. Similarly, we found that extremely high CCMAS measures did not indicate the item was selective *exclusively* to one category as might be expected. To take an extreme example, we found a unit in the output prob layer that had selectivities of .999 for category 'bridegroom' and .995 for category 'flowerpot' . Top-class selectivity for this unit was different once again, responding most strongly to the category 'ringlet butterfly'.

At the same time, we identified problems with the localist, top-class, and activation maximization (AM) methods as well. The localist selectivity measure failed to obtain any localist representations, even at the output prob layer of AlexNet. This measure is so extreme that it misses highly selective representations that are of theoretical interest. The top-class selectivity does provide a graded measure of selectivity (with 100% top-class selectivity equivalent to a localist grandmother cell), but it can underestimate selectivity when a few member from outside the top-class are highly activated (see Fig. 4b for an example). At the same time, the human interpretation of AM images provides a poor measure of hidden-unit selectivity given that interpretable AM images were associated with low top-class selectivity scores. These findings highlight the need to provide better measures of selectivity in order to better characterize the learned representations in NNs.

What should be made of the contrasting findings that localist representations are found in RNNs, but not in AlexNet? The failure to observe localist units in the hidden layers of AlexNet is consistent with the Bowers et al. 2014 claim that these units only emerge in order to support the co-activation of multiple items at the same time in short-term memory. That is, highly selective representations may be the solution to the superposition catastrophe, and AlexNet only has to identify one image at a time. This may help explain the many reports of highly selective neurons in cortex given that the cortex needs to co-activate multiple items at the same time in order to support short-term memory (Bowers et al., 2016).

However, it should be noted that the RNNs that learned localist units were very small in scale compared to AlexNet, and accordingly, it will be interesting to assess unit selectivity in larger RNNs that have much larger memory capacity. Relevant to this issue, Karpathy et al. (2016) reported some striking examples of selective representations in a RNN with long-short term memory (LSTM) trained to predict text. Although they did not systematically assess the degree of selectivity, they reported examples that are consistent with 100% selective units. If in fact the superposition constraint provides a pressure to learn more selective representations, then we should observe more highly selective representations, perhaps localist units, in large RNNs with LSTMs as well. We will be testing this hypothesis in future work.

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

# APPENDIX FOR SELECTIVITY METRICS CAN OVERESTIMATE THE SELECTIVITY OF UNITS: A CASE STUDY ON ALEXNET

## A1    METHODOLOGICAL DETAILS FOR THE BEHAVIORAL EXPERIMENT

One hundred generated images were made for every unit in layers conv5, fc6 and fc8 in AlexNet, as in Nguyen et al. (2017), and displayed as 10x10 image panels (figures A7 and Figures A5 and A6). A total of 3,299 image panels were used in the experiment (995 fc8, 256 conv5, and 2048 randomly selected fc6 image panels) and were divided into 64 counterbalanced lists of 51 or 52 (4 conv5, 15 or 16 fc8 and 32 fc6). 51 of the lists were assigned 5 participants and 13 lists were assigned 6 participants.

To test the interpretability for these units as object detectors, paid volunteers were asked to look at image panels and asked if the images had an object / animal or place in common. If the answer was 'yes', they were asked to name that object simply (i.e. fish rather than goldfish). Analyses of common responses was done for any units where over 80% of humans agreed there was an object present, by reading the human responses and comparing them to both each other and to the output classes. Agreement was taken if the object was the same rough class. For example, 'beer', 'glass', and 'drink' were all considered to be in agreement in the general object of 'drink', and in agreement with both the classes of 'wine glass' and 'beer' as these classes were also general drink classes (this is an actual example, most responses were more obvious and required far less interpretation than that). Participants were given six practice trials, each with panels of 20 images before starting the main experiment. Practice trials included images that varied in their interpretability.

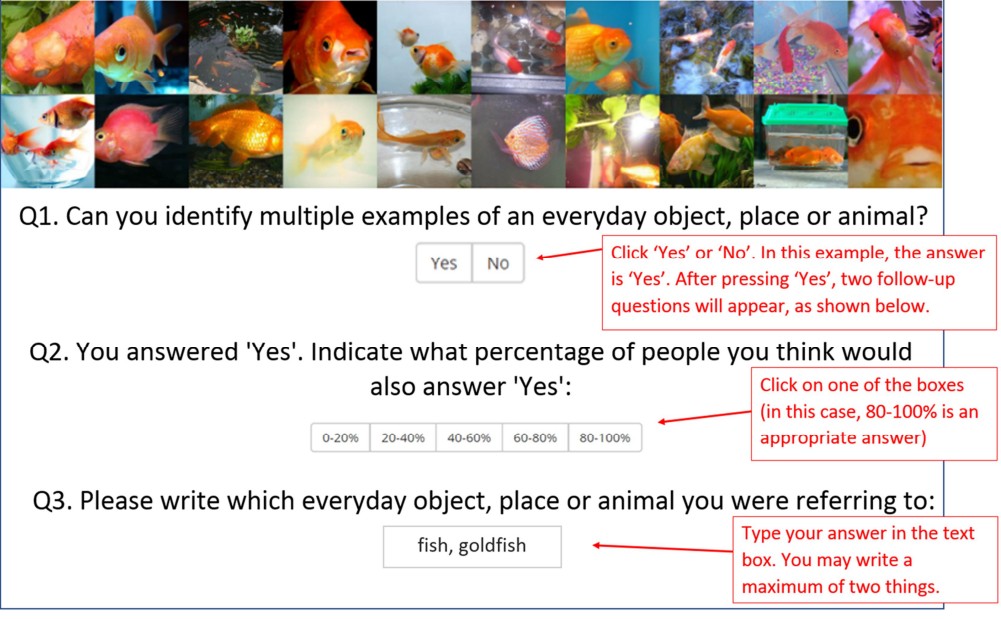

Figure A1: Example screen from the identification task shown to participants as part of the instructions. The images included on this practice trial are ImageNet2010 images, not AM images.

## A2 FURTHER DATA ON THE SELECTIVITY MEASURES

Further data on the selectivity measures across AlexNet. Table A1 demonstrates some extreme values of CCMAS, precision, and top-class selectivity as well as the similarity between the CCMAS ad CCMAS_2, and Table A2 gives the mean values across the layers.

Table A1: CCMAS selectivity measures for extreme example units. Across all the tested units, prob.322, fc8.393, fc7.31, fc6.582 and conv5.unit78 were the units with the highest CCMAS measures. Units fc6.1199, fc8.11 and prob.11 were displayed in Figs. 3 & 4.

| LAYER.UNIT | CCMAS | CCMAS_2 | Precision | Top Cluster Sizes | % Top Class Selectivity |
|---|---|---|---|---|---|
| **Top CCMAS units** | | | | | |
| prob.322 | 0.999921 | 0.995384 | 100% | 963 | 97.1% |
| fc8.393 | 1.24357 | 1.39617 | 95% | 29 | 3.3% |
| fc7.31 | 0.936767 | 0.865305 | 11% | 1 | 0.15% |
| fc6.582 | 0.933832 | 0.919425 | 1% | 1 | 0.14% |
| conv5.78 | 0.746763 | 0.746346 | 5% | 1 | 0.10% |
| **Top precision units** | | | | | |
| prob.0 | 0.9996680 | 0.99316330 | 100% | 1000 | 92.2% |
| fc8.1 | 1.049710 | 1.110110 | 100% | 172 | 16.1% |
| fc7.255 | 0.8961654 | 0.84108156 | 97% | 94 | 7.6% |
| fc6.1199 | 0.92323 | 0.818260 | 95% | 43 | 3.5% |
| conv5.0 | 0.554049430 | 0.528534300 | 77% | 1 | 0.1% |
| **Top class selectivity units** | | | | | |
| prob.322 | 0.99964017 | 0.99538374 | 100% | 963 | 97.1% |
| fc8.985 | 1.0908700 | 1.1929800 | 100% | 574 | 50% |
| fc7.255 | 0.8961654 | 0.84108156 | 97% | 94 | 7.6% |
| fc6.1199 | 0.92323 | 0.818260 | 95% | 43 | 3.5% |
| conv5.100 | 0.68313890 | 0.6831976 | 56% | 9 | 1.1% |
| **Example units** | | | | | |
| prob.11 | 0.999876 | 0.973243 | 100% | 1000 | 95.2% |
| fc8.11 | 1.10469 | 1.1946 | 99% | 88 | 8.4% |

Table A2: Average CCMAS measures across layers (note that these values are not normally distributed, see figures A2 and A3.

| LAYER.UNIT | CCMAS | CCMAS_2 | Precision | Top Cluster Sizes | % Top Class Selectivity |
|---|---|---|---|---|---|
| Mean[prob] | 0.999309 | 0.984169 | 99.7% | 592.5 | 82.1% |
| Mean[fc8] | 0.995644 | 1.002905 | 70.1% | 108.4 | 5.1% |
| Mean[fc7] | 0.847799 | 0.830854 | 15.4% | 20.3 | 0.23% |
| Mean[fc6] | 0.8439226 | 0.821118 | 12.1% | 17.4 | 0.19% |
| Mean[conv5] | 0.491029 | 0.464662 | 9.6% | 17.3 | 0.17% |

## A3 FURTHER ISSUES WITH THE CCMAS MEASURE

### A3.1 HISTORGRAMS AND DISTRIBUTION FITS FOR ACTIVATIONS IN UNIT fc6.1199

The CCMAS measure is based on comparing the mean activation of categories, and this is problematic for a few reasons. First, the majority of images do not activate a unit at all. For instance, our butterfly 'detector' unit fc6.1199 has 82.8% of images with an activation of 0.0 (see figure A2). This means that the CCMAS selectivity is largely determined by the the distribution of images that have 0 activation rather than the distribution of images that strongly activate a unit. This leads to very different estimates of precision, top-class and localist selectivity that are concerned with the distribution of highly activated units. Note that, this issue with activations of 0.0 moving the mean around could be solved by taking only the non-zero activations (which is not that dissimilar from what neuroscientists do), however there are problems with the non-zero activations.

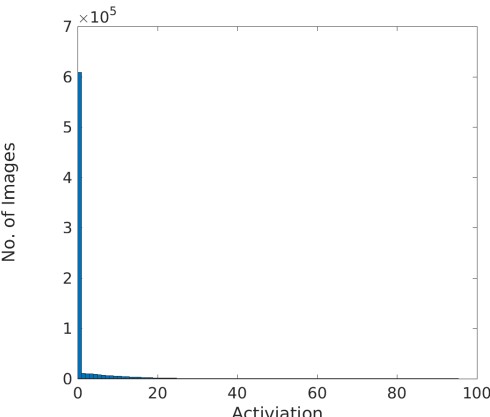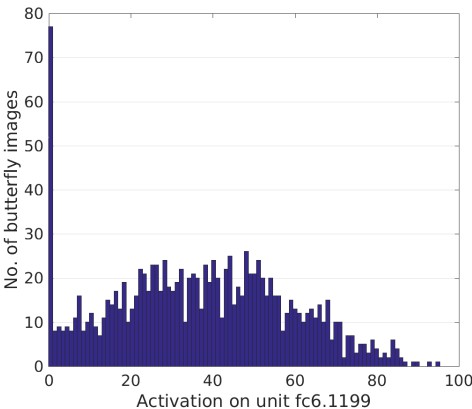

Figure A2: The distribution of activations on unit fc6.1199 for all images (left). 82.8% of activations are zero across all classes, only 5.8% of the butterfly class are zero. Unit fc6.1199 is a butterfly detector under Zhou et la's precision measure. Bins are 1.0 wide.

The problem with the CCMAS measure when applied to the non-zero activations is that they are not normally distributed. As figure A3 demonstrates (for our example unit fc6.1199), the all non-zero activations follow an experiential curve and thus the mean is not a useful measure. Fitting a normal distribution to this data gives the blue normal distribution curve. Although the butterfly class can be roughly fit by a normal distribution, as the entire activations follow an exponential, the non-butterfly classes will be best fit by an exponential not a normal distribution. As the CCMAS requires a comparison of means, and the not-A classes follow an exponential, rather than normal distribution, it follows that the CCMAS will give misleading results.

Computing CCMAS on the basis of mean activations can produce highly non-intuitive as illustrated in figure A4 that plots three distributions of generated data from 10 classes of 100 points. We can see that the CCMAS gives the same (and maximal) score for the case where the unit is off to everything but a single point (figure A4a) as it does for a 'grandma unit' (see figure A4c). And the CCMAS gives an incredably low score when the means of A and not-A are similar, even if they perfectly separate the disjoint sets (figure A4b).

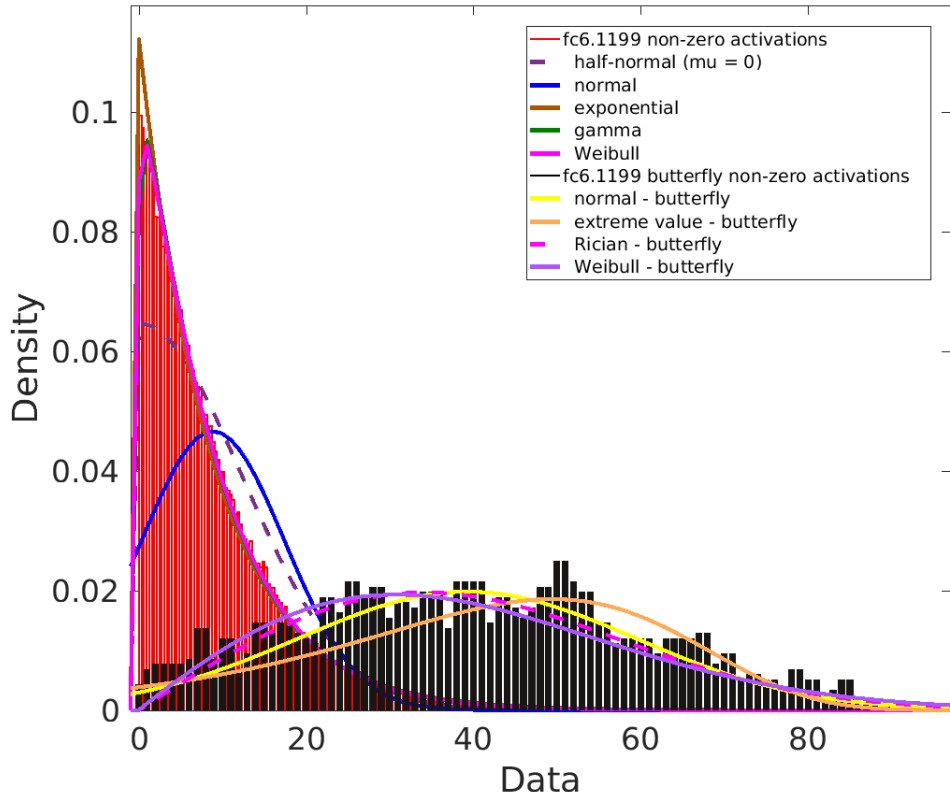

Figure A3: Fitting the non-zero activations for all classes (red) and the maximum activation class (black). For the superset of all the classes, the distribution is well-described by exponential-derived fits, normal-derived fits are bad. For the maximum activating class (butterfly), the distribution has a mean and can be well-described as a normal distribution.

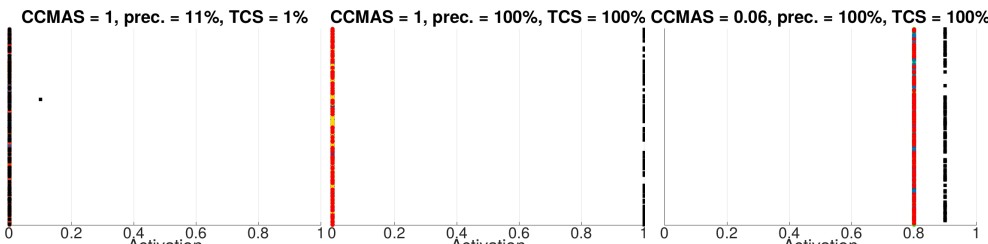

(a) One active item from one class. CCMAS = 1, precision = 11%, TCS = 1%.

(b) Archetypal 'grandmother' unit. CCMAS = 1, precision = 100%, TCS = 100%.

(c) One class more active than the others. CCMAS = 0.06, precision = 100%, TCS = 100%.

Figure A4: Example of where the CCMAS does not match intuitive understandings of selectivity. Generated example data: (a) If a unit is off to all but a single image from a large class of objects, the CCMAS for that class is 1 (maximum possible selectivity). (b) If a unit is strongly activated to all members of one class and off to everything else (an archetypal 'grandmother' cell) the CCMAS is the same as for (a) although the precision and top-class selectivity is vastly different. (f): If a unit has high activations for all classes, but one class (black squares) is 0.1 more than all others (coloured circles), the CCMAS is very low (0.06) despite being %100 top-class selective. The generated examples are for 10 classes of 100 items

## A4 HUMAN INTERPRETATION OF ACTIVATION MAXIMIZATION IMAGES

Some examples of the 10x10 grids of activation maximisation images that were presented to participants are shown in Figures A5, A6 and A7. Figure A5 shows an example from conv5 that human participants agreed had no obvious object in common (although there are repeated shape motifs, the participants were specifically asked for objects, and not abstract concepts like shape or color. Figure A6 is also from the conv5 and was judged by participants as some images containing 'dogs'. Figure A7 is the AM images for the supposed 'butterfly detector' unit example discussed in the paper.

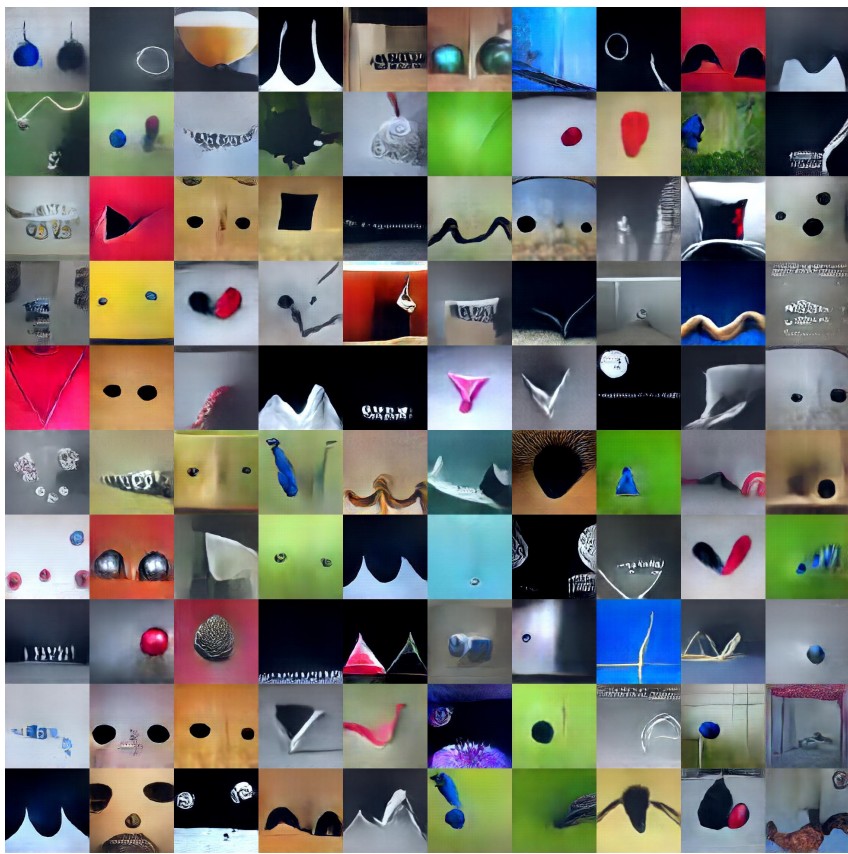

Figure A5: Example activation maximisation images for unit conv5.65. These images were judged by humans to not contain any interpretable objects in common (although the reader may agree that there are some shape and colour similarities in the images).

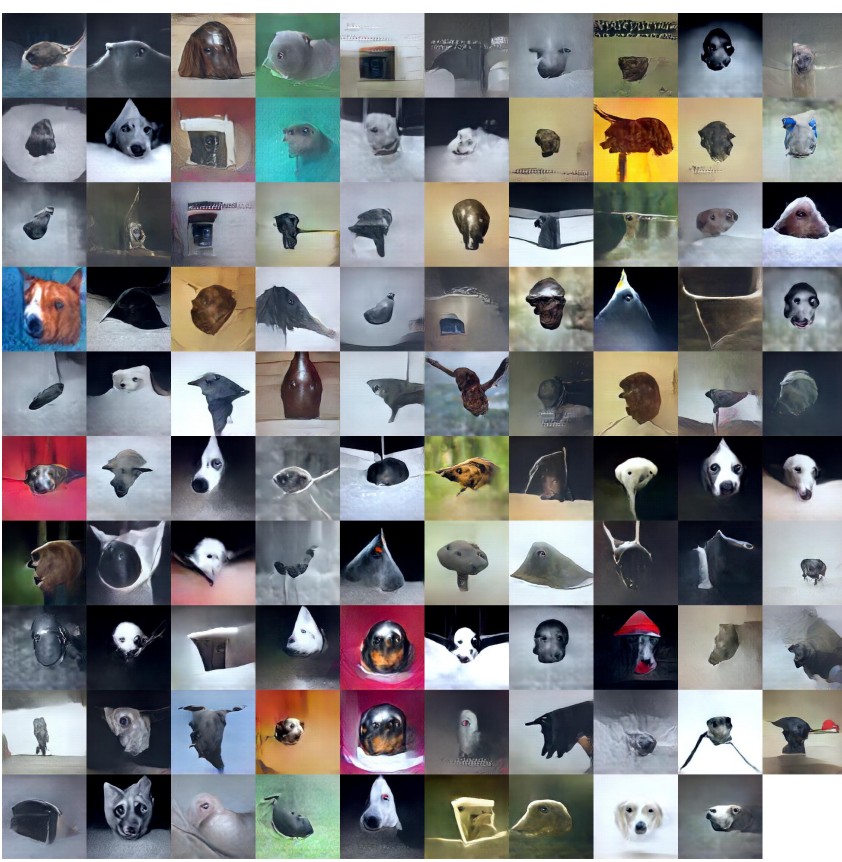

Figure A6: Example activation maximisation images for unit conv5.183. These images were judged by humans to contain some interpretable images, in this case, of the type 'dogs'.

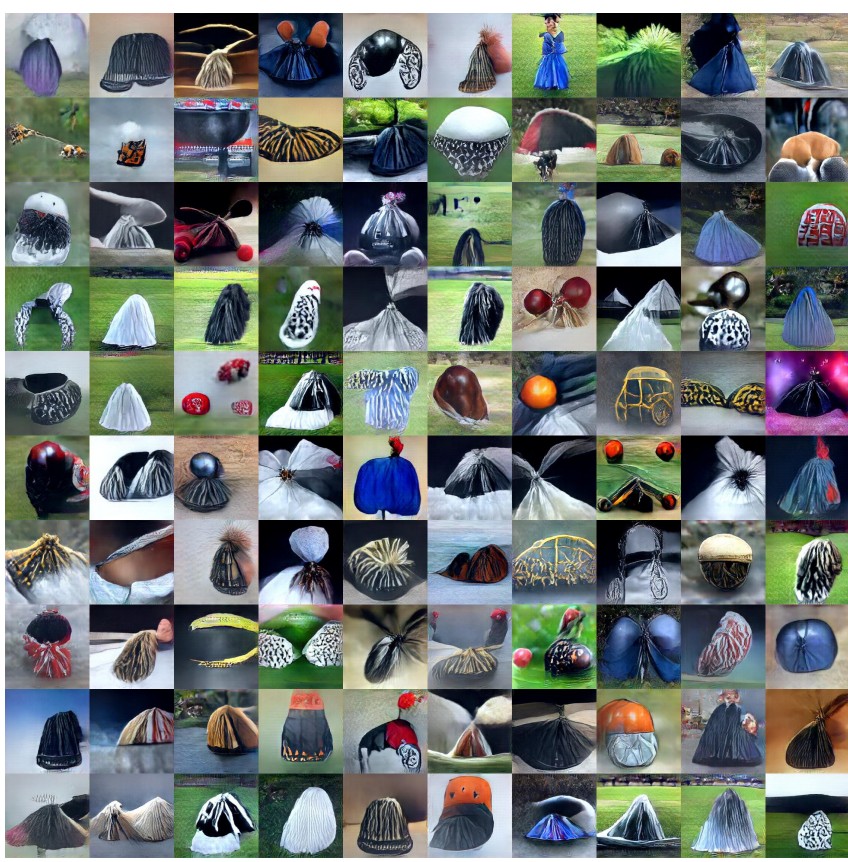

Figure A7: Example activation maximisation images for unit fc6.1199. Whilst there are some butterfly wing shapes in these images, there are not obvious butterflies. N.B. the second highest activating class for this unit is ladybirds, and there are some orange round shapes that could conceivably be ladybug-alikes.

