# OpenReview forum: "Selectivity metrics can overestimate the selectivity of units: a case study on AlexNet"
_ICLR.cc/2019/Conference_

### Official Review · AnonReviewer3 · 2018-11-02
**Poorly written with unclear contributions**

**Rating:** 3
**Confidence:** 5

**Review:**

This is a paper with scattered potentially interesting ideas. But the execution is limited and the writing poor with critical details lacking.  A major limitation of the paper is that it is not clear what contribution it makes. Some of the analyses are indeed interesting but 1) these analyses are mostly descriptive and 2) they are limited to one particular (outdated) architecture. How would batch norm or residual connections or any of the developments that have happened since AlexNet affect these results?

As a side note, the references/comparisons between AlexNet and recurrent nets (see abstract, etc) are misleading. This is based on the claim that Bowers et al (2014) qualitatively different results but this is for entirely different domains (words). Indeed what could have made potentially the work more relevant would have been to show some kind of benchmarking between AlexNet and alternative architectures (possibly RNNs). As such the current study does not contribute much except for comparing different semi-arbitrary measures of selectivity for one specific (outdated) network architecture trained on a particular problem (ILSVRC).

****
Minor points:

The study is limited to correctly classified images as stated on page 3. This seems like a major confound in a study aimed at understanding the visual representations learned. It seems to me that the conclusions of the paper could be heavily biased because of this (when computing any measure based on inter and intraclass responses).

In general, this is a relatively poorly written paper which would be hard to reproduce. For instance, the image generation for activating units (assuming it is novel) could be interesting but it is not even described with sufficient details so as to reproduce the results.

---

> ### Author Response · Authors · 2018-11-26
> **Clarifying our contribution - part 2**
>
> >> R3 said that our results would be hard to reproduce.
>
> Response: The procedures for computing precision and CCMAS measures have been published elsewhere and cited (and we explain how they are computed), and we have provided the equation for computing top-class selectivity.   We also cite the paper that describes how to generate the images that maximally activated the units. In order to facilitate replication we have uploaded a file that contains all the activations of correctly identified images as an h5 file and provide a link in the paper.
>
> >> R3 said the image generation for activating units (assuming it is novel) could be interesting but it is not even described with sufficient details so as to reproduce the results.
>
> Response: The process of generating Activation Maximization (AM) images follow exactly a state-of-the-art AM method in Nguyen et al. 2017 via their open-source code. We thank the reviewer for pointing this out! :)
> We have updated the manuscript to add more details and make this clearer.
>
> >> R3 said that ‘The study is limited to correctly classified images as stated on page 3. This seems like a major confound in a study aimed at understanding the visual representations learned.’
>
> Response: We do not understand this point.  It is not appropriate to compute the selectivity of a unit when misclassified images are included.  For the sake of argument, imagine you have found a unit that appears to be 100% selective to DOGS.   What should you conclude if the unit does not activate to an image of a DOG that is misclassified as a CAT.  Should you conclude that it is not a 100% DOG detector? Of course not – the DOG detector did not fire because the model did not know it was a DOG.  If you are interested in whether a unit is selective to a given category the model needs to correctly identify the category.   It is an interesting question as to why models sometimes misclassify images, but goes beyond the topic of this paper.

---

> ### Author Response · Authors · 2018-11-26
> **Clarifying our contribution - part 1**
>
> Please see the main reply to your major points. Below is our reply to your specific comments.
>
> >> R3 does not like the jitterplot method of displaying the results.
>
> Response: We find this an intuitive method to display the data.  The fact that some regions of the scatter plot have highly overlapping dots is itself informative as it shows that there is no selectivity in this range of activity.  But we have added a histogram to some of our jitterplots so the reader can see how the lower activations are laid out (See Figures 3 and A2 ).
>
> >> R3 writes that our analyses are mostly “descriptive”.
>
> Response: We do not understand this point.  We have compared four quantitative measures of selectivity (localist, top-class, CCMAS, and precision selectivity), and for the first time we have provided quantitative measures of the selectivity from the activation maximization method.
>
> >> R3 wonders how batch normalization would impact our results.
>
> Response: Again, this goes beyond the main focus of our paper.  No doubt batch normalization would impact on all our measures (as would dropout, L1 and L2 regularization, using different network architectures, etc.), BUT the discrepancy between measures would remain (this is our main point).
>
> >> R3 claims the comparison between AlexNet and recurrent nets are misleading given that Bowers et al. (2014) worked with words.
>
> Response: It is important to note that Bowers (2014, 2016) did not find selective representations for words when the networks were trained on words one-at-a-time, but only when trained to co-activate multiple words at the same time in short-term memory (see Figure 1b).  That is, the critical contrast here is not in words vs. images, but in training conditions.  Bowers et al. (2014, 2016) argued that selective units developed in response to the “superposition catastrophe” that limits the computation capacities of distributed representations.
> With regards to whether we were being “misleading”, we would point out that we explicitly noted that it would be important to determine whether more recent recurrent CNNs would also learn selective representation as predicted on the superposition account.  Specifically, we wrote:
> “However, it should be noted that the RNNs that learned localist units were very small in scale compared to AlexNet, and accordingly, it will be interesting to assess the selectivity in larger RNNs that have much larger memory capacity.  Relevant to this issue, Karpathy et al. (2015) reported some striking examples of selective representations in a recurrent long-short term memory (LSTM) networks trained to predict text based on training on Tolstoy's novel `War and Peace' and the Linux Kernel source code. Although they did not systematically assess the degree of selectivity, they reported examples that are consistent with 100% selective units. If in fact the superposition constraint provides a pressure to learn more selective representations, then we should observe more highly selective representations, perhaps localist units, in large RNNs as well.  We will be testing this hypothesis in future work.”

---

### Official Review · AnonReviewer2 · 2018-11-02
**Surprising result and raises interesting questions.**

**Rating:** 6
**Confidence:** 3

**Review:**

Summary - This paper analyzes the selectivity of individual units in CNNs. The authors analyze existing techniques such as precision selectivity, class-conditional mean activity selection and localist sensitivity. These methods are analyzed in the context of AlexNet and ImageNet. The authors also use Activation Maximization (AM) techniques for visualizing single-unit representations in CNNs.


Paper strengths
- The authors have minutely examined each of the metrics and the underlying assumptions they make.
    - Example - Number of images used for computing the precision threshold in Zhou et al., 2014; The wrongly stated range of CCMAS [0, 1]. Considering the second highest CCMAS class is a good way of handling multiple classes that activate a single unit.
- The results of this paper are surprising compared to existing work. The authors have made a surprising discovery and done a good job of both presenting it well and experimentally validating it. The paper raises interesting questions and this should inspire future work in understanding networks.
- Figure 2 is insightful - It compares the various different interpretations of selectivity for a single unit in fc6. It shows how the mean activating class and the maximally activating class can be semantically very different. It also shows that despite the high precision and CCMAS score, the unit cannot be labelled as a detector for the single concept "custard apple". More such results are presented in the Appendix (e.g. Fig A6)
- The human study in Section 3.3 is a good way to evaluate the generated AM images.


Paper weaknesses
- One of the major weaknesses of this paper is that it uses only ImageNet images to evaluate the units. As this is limited to 1000 classes, the authors cannot probe other visual concepts such as color, texture, materials for the units. As an example, Network dissection (Zhou et al., 2017) proposes a dataset called Broden which has many diverse sets of visual concepts labeled. This paper focuses only on one definition of selectivity - selecting objects. This should be made explicit and the authors have not done a good job of clarifying this assumption or showing that it exists.
- All of the analysis is limited to AlexNet. With modern architectures that use residual/skip connections, it is not clear how well this analysis will generalize. It is an open question if the authors work overfits to AlexNet.
- The jitterplots are hard to understand especially if there are many overlapping "dots" (samples). Since the y-axis values are not really meaningful anyway, using a histogram to see how many samples have a particular activation value is easier. A possible suggestion for Figure 2(a): split into two parts - 1) histogram of all samples; 2) histogram of the highest mean activating class.
- The organization of the paper could be improved. The sections in the paper are not well connected.

---

> ### Author Response · Authors · 2018-11-26
> **We clarified that we are interested in object selectivity**
>
> We thank R2 for their constructive comments and encouragement! Please see the main reply to all reviewers for the main points. Below is our reply to your specific comments.
>
> >> R2 points out that we are only concerned with object selectivity whereas Zhou et al. (2017) are concerned with different forms of selectivity, including selectivity to colour, etc.
>
> Response: This is true, and we have now emphasized that the focus of this paper is on object selectivity.  There may well have been units selective to colour that our analyses may have missed, but this does not impact on our conclusion that the precision measure provides a misleading measure of selectivity (not only misleading for objects, but for the same reason, misleading for colour, etc.).
>
> >> R2 said: It is an open question if the authors work overfits to AlexNet.
>
> Response:  The method of comparing selectivity measures could be applied to any NN, and AlexNet's activation patterns are similar to other NNs, so we think another NN would also show similar qualitative pattern of low object selectivity in the hidden layers, and we still expect the selectivity metrics to underestimate this. Our preliminary results with other networks do look similar. Where AlexNet differs, perhaps, is in the quality and style of AM images that are found, so that part of our analysis may be different, and this is something we intend to investigate in the future.

---

### Official Review · AnonReviewer1 · 2018-11-03
**Shows that single units are not perfectly class selective, a result that will be intuitive to many**

**Rating:** 5
**Confidence:** 3

**Review:**

Summary:

This paper explores different metrics to measure the ‘selectivity’ of single neurons for a class in deep neural networks. Using AlexNet as the model under study, the paper shows strengths and weaknesses of several recent methods in the literature. The paper conducts a psychophysics experiment to see if human subjects can reliably label images generated through activation maximization techniques.

Major comments:

This paper undertakes a careful analysis of different ways of measuring single-unit selectivity for a class. The conclusions drawn are that no neurons exhibit true localist selectivity, and most have some more complex selectivity. Some of the specific examples make this point very nicely (for instance, a unit that responds very strongly to several custard apples and would appear to be a custard apple detector, except that it responds extremely weakly to other custard apples). This is a somewhat negative result that may be useful in advancing the field away from single neuron analyses, which may be misleading.

One worry is that the methods applied are looking for a very strong form of selectivity. In particular, even the output layer is judged to contain a low percentage of selective units according to the definitions in the paper. It may be worth considering slightly weakened versions of the metrics that allow for some errors.

It would be useful to add discussion of the connections between these metrics and generalization performance. The class conditional selectivity metric, for instance, may not measure localist coding very directly, but it does correlate with important performance metrics like generalization performance. The discussion in Morcos 2018 suggests that high single unit selectivity is detrimental to generalization. Do these correlations persist using other metrics?

The psychophysics experiment with human subjects appears to have been done to a high standard, and yields the result that only the very highest layers of a network yield interpretable images. This is somewhat interesting but unlikely to be that surprising, as selectivity for objects in lower layers is not a claim made by many works. In these lower layers, selectivity for ‘object parts’ is a claim that has been made and could potentially be addressed by the data collected.

Overall this paper critically analyzes single unit selectivity measures, reaching the conclusion that tuning in modern deep networks is usually far more complex than strict localist coding. The significance of this conclusion may not be so high given that this conclusion is probably already the intuition of many.

---

> ### Author Response · Authors · 2018-11-26
> **Not only are units not perfectly class selective, we show that the selectivity measures previously introduced are misleading as to how unselective they are.**
>
> We thank R1 for their constructive comments and encouragement!  Please see the main reply to your general, main points. Below is our reply to your specific comments.
>
> >> R1 would like us to assess the role of selectivity and generalization as shown in Morcos (2018).
>
> Response: In fact, Morcos et al. did not show this (they showed that generalization related to a concept called “single directions”, but they never associated their selectivity measure CCMAS to generalization).  We agree this is an interesting question, but it goes outside the scope of comparing selectivity measures.  We would note that the Bowers (2016) paper was specifically concerned with the issue of generalization and selectivity in the context of a recurrent network, and in that paper, selectivity was required in order to generalize.  In future work, it would be interesting to compare how selectivity and generalization relate in feedforward and recurrent networks.
>
> >> R1: Overall this paper critically analyzes single unit selectivity measures, reaching the conclusion that tuning in modern deep networks is usually far more complex than strict localist coding. The significance of this conclusion may not be so high given that this conclusion is probably already the intuition of many.
>
> Response: We agree that the fact that we did not find 100%-selective units in AlexNet might not too surprising, but that is only one contribution. The main and most important contribution in our work is that existing measures of selectivity in established literature can be misleading. That is impeding the community’s understanding and advancement of the Interpretability research.  As noted above, our findings also highlight an interesting question for future research, namely, what are the conditions that do lead to 100% selectivity, as observed by Bowers et al. (2014, 2016).

---

### Author Response · Authors · 2018-11-26
**Our response to all reviewers - part 2: we rewrote and reorganised the paper for clarity**

4:  Paper was not well written or well organized.

Response:  We have substantially rewritten the paper to clarify our objectives, and to improve the organization of the results.   We have also updated the manuscript with more details to reproduce our findings, including providing a link to h5 file that includes the activations of the units in AlexNet, and provided more details regarding Activation Maximization, which followed the released code by Nguyen et al. (2017).

An aside
We have also identified a bug in our code that altered some of our findings (we discuss this under Point 5). However, the bug does not change our main conclusions, indeed, the results are now stronger.

We identified a bug in our code that artificially lead to higher levels of selectivity according to our top-class selectivity measure.  By contrast, the effects on precision and CCMAS were not as strongly impacted.  As a consequence the differences between measures has been increased now that we have fixed our code (strengthening our main message).  We have updated tables and figures in response to our mistake.  Briefly, the bug involved copying around a pointer to a memory address containing data, rather than the data itself, leading to some activations not being included in the calculation.  The qualitative findings of the paper have not changed, but the precise values have.
In sum, we think we have made a number of important contributions,  and we hope that our comments here and in our revision highlight this better.

---

### Author Response · Authors · 2018-11-26
**Our response to all reviewers - part 1: AlexNet was used by the people developing these selectivity measures, and so a good case study**

We would like to thank the reviewers for their positive comments and their constructive criticisms.   Below we respond to these criticisms, some of which reflect a misunderstanding of our our main objectives (this is our fault--and something we clarify here and in our revision).  We hope our revision makes it clearer that our paper has identified serious problems with influential and recent papers published in ICLR that overestimate the selectivity of units in CNNs. Here we consider the points that are relevant to all the reviewers, and then respond to the specific comments of the reviewers below their reviews.

1:  We have only used AlexNet, and accordingly, it is not clear how well our findings will generalize to other more recent networks.

Response:  The main goal of our paper was to compare different measures of selectivity, and for this purpose, it makes sense to focus on a single network.

(1) Considering alternative models would NOT alter our finding that the different selectivity measures assess very different things.

(2) We decided to use AlexNet given its historical significance and the fact that many papers have explored the selectivity of the hidden units in this network (in fact, two of the three papers which introduced the selectivity measures we’re testing used AlexNet, the third paper used a NN on a similar size to AlexNet, thus AlexNet is the best network for directly comparing our findings to the original papers).  As described in the title of our paper, we used AlexNet as a “case study” for this purpose.

2:  We are looking for a very strong form of selectivity, and the fact that we do not find 100% selective “grandmother cell” representations is not that surprising.

Response:

(1) We are not looking for a strong form of selectivity, but rather, we are comparing different measures of selectivity.  Our comparison highlights that the words “selective”, “detector” and “precision” are being used in somewhat misleading ways.  eg,, Zhou et al reported multiple “object detectors” in AlexNet with a precision of above .75, ie.  75% of the top-most active images are members of a single object category. This was published in ICLR, and is highly cited (384 in Google Scholar). By contrast, we show that these object detectors do not strongly respond to the vast majority of images from the category, with the modal response to images from these categories often 0.  Indeed, we did find many units in AlexNet that should NOT be called “object detectors”.  We show that all current selectivity measures have their problems, and that new measures are needed.

(2) We agree that there are no 100% grandmother cell representations in AlexNet might not be too surprising, but it is surprising that when we use Morcos et al’s CCMAS measure (ICLR 2018) we find hidden units with a selectivity score of .94 that would appear to suggest near localist codes (whereas the  top-class selectivity scores for the same unit is .15%).

(3) It is also important to note that 100% selective units have been found in recurrent networks (Bowers, 2014, 2016), and this raises interesting questions for future research regarding the conditions in which 100% selective units are found.

3:  It is not clear what the contribution is.

Response: There are a number of contributions:

(1) The paper compares for the first-time four different measures of selectivity: (a) localist selectivity (from psychology; Berkeley et al., 1995, Bowers, 2014); (b) Top-Class selectivity, with “localist” selectivity as a special case of 100% top-class selectivity  ; (c) Precision (Zhou et al. 2015); (d) CCMAS (Morcos et al. 2018). In addition, we evaluate the images by a 4th popular method (e) Activation Maximization (Erhan et al. 2009, Simonyan et al. 2014, Yosinski et al. 2015, Nguyen et al. 2014-2017etc).

We show that they assess very different things that lead to very different conclusions regarding the selectivity of object information in AlexNet.

(2) The paper introduces the “localist” measure of selectivity to the machine learning community, which has been used in top psychology journals, but is not reference in the machine learning literature.

(3) Our findings motivate  new areas of research, including questions concerning why 100% selective units have thus far only been observed in recurrent networks (perhaps the “superposition catastrophe” is central here, but more research is needed), and the need for new and perhaps multiple measures of selectivity given the limitations of all current measures.

---

### Meta-Review · Area_Chair1 · 2018-12-14
**Detailed analysis of unit selectivity, but reviewers unconvinced of impact**

**Confidence:** 4
**Recommendation:** Reject

**Metareview:**

The paper examined the folk-knowledge that there are highly selective units in popular CNN architectures, and performed a detailed analysis of recent measures of unit selectivity, as well as introducing a novel one. The finding that units are not extremely selective in CNNs was intriguing to some (not all) reviewers. Further, they show recent measures of selectivity dramatically over-estimate selectivity.

There was not tight agreement amongst the reviewers on the paper's rating, but it trended towards rejection. Weaknesses highlighted by reviewers include lack of visual clarity in their demonstrations, the use of a several-generations-old CNN architecture, as well as a lack of enthusiasm for the findings.